# Accuracy and sensitivity of $NH_3$ measurements using the Dräger Tube Method

Alexander Kelsch[1], Matthias Claß[1], and Nicolas Brüggemann[1]

[1]Forschungszentrum Jülich GmbH, Institute of Bio- and Geosciences – Agrosphere (IBG-3), Jülich, 52428, Germany

**Correspondence:** Alexander Kelsch (a.kelsch@fz-juelich.de)

**Abstract.** Regional estimates of ammonia ($NH_3$) emissions are often missing data from heterogeneous or small fields. Areas with no experienced staff or in-field power supply also prevent the use of accurate and fully established micrometeorological measurement techniques. The Dräger Tube Method (DTM) is a calibrated open-dynamic chamber method, which requires little training to use and is relatively inexpensive. It uses $NH_3$ detector tubes (Dräger Tubes), an automatic pump, as well as a chamber system comprised of four stainless steel chambers connected with PTFE tubing. Even though the DTM is often used in countries such as Germany and China, the detection accuracy, precision and sensitivity have not been tested yet. In order to quantify those for the DTM, we simultaneously measured defined $NH_3$ mixing ratios with the Dräger Tubes, with quantum cascade laser spectroscopy (QCLS) (MGA[7], MIRO Analytical AG, Switzerland) and with cavity ring-down spectroscopy (G2103, Picarro, Inc., USA). Second, we tested the exchange of the tubing material and heating of the tubing under laboratory conditions, as well as PTFE film attachments or wiping of the DTM chamber system with ethanol during outdoor measurements, on performance improvements. Results showed that the Dräger Tubes had a detection limit between 150 and 200 ppb, which is three to four times higher than originally assumed. Dräger Tube concentration measurements also underestimated $NH_3$ concentrations by 43 to 100 % for mixing ratios between 50 and 300 ppb, and by 28 to 46 % for mixing ratios between 500 and 1500 ppb. The PTFE tubing material showed similar performance to the polyester-polyurethane tubing material regarding response time, which was further improved by heating the tubing to 50 °C. The modifications of the chamber surface and cleaning in the outdoor experiment did not lead to any improvements of $NH_3$ concentration measurements. The results suggest that the DTM should only be used where alternatives are unfeasible and high $NH_3$ emissions are to be expected. Further assessment of calibrated DTM using reference methods is required for a comprehensive evaluation and alternative developments for a more appropriate method replacing the DTM in small plot applications is encouraged.

## 1 Introduction

Ammonia ($NH_3$) is one of the main air pollutants in Europe (European Environment Agency, 2023). Volatilization of $NH_3$ from agriculture is by far the largest source of anthropogenic $NH_3$ emissions and is responsible for 94 % of emissions in the European Union (European Environment Agency, 2023). $NH_3$ is highly reactive and combines with other molecules in the atmosphere such as sulphuric acid, nitric acid or hydrochloric acid to form particulate matter less than 2.5 $\mu$m in size, which has been shown to cause premature death, respiratory infections and diseases, lung cancer and cerebrovascular disease (Lelieveld et al.,

2015; Lim et al., 2012; Wang et al., 2017). Most volatilized $NH_3$ is transported by wind and deposited on the Earth's surface, either dissolved in water through wet deposition or attached to other particulate matter through dry deposition (Cameron et al., 2013). Deposition in aquatic and terrestrial ecosystems can lead to eutrophication and acidification, which has been shown to result in biodiversity loss (Behera et al., 2013). In addition, the volatilization of $NH_3$ causes indirect greenhouse gas emissions once it is partially converted to nitrous oxide through bacterial nitrification after re-deposition into the soil. The reduction potential of $NH_3$ in the EU is 20–35 % compared to year 2000 emission levels, and the environmental, health and economic benefits (including the reduced need for fertilizers) far exceed the necessary reduction costs (Zhang et al., 2020). Therefore, the NEC Directive 2016/2284 requires the EU member states to reduce their total $NH_3$ emissions by 19 % by 2030 compared to year 2005 levels (EU Directive, 2016). To reduce the impact of agriculture on $NH_3$ volatilization, it is crucial to accurately quantify emissions from various types of fertilizers and evaluate effective mitigation options. $NH_3$ emissions can be measured on a global scale using satellite observations, such as those from the Cross-track Infrared Sounder (CrIS) or the Infrared Atmospheric Sounding Interferometer (IASI), these satellite-derived estimates still rely on field data for validation to ensure their accuracy and applicability to different environments. The lack of field measurement data where micrometeorological methods could be impractical due to smaller and heterogeneous plot sizes is mentioned as a major uncertainty (Behera et al., 2013; Dammers et al., 2019; Luo et al., 2022).

There are various approaches to quantifying $NH_3$ emissions in the field. The most common are micrometeorological and chamber methods (Di Perta et al., 2020). Micrometeorological methods are considered to be the most accurate. However, they are unsuitable for comparisons between several plots close to each other or for smaller, heterogeneous fields and those without power supply (Pacholski et al., 2006; Roelcke et al., 2002). Chamber methods operate on the principle that $NH_3$ volatilizes into a hood placed over the emitting soil for a defined period.

Chambers can be broadly divided into static chambers, where there is no forced air circulation, and dynamic chambers, where there is forced air circulation using, for example, a pump or fan. In addition, both static as well as dynamic chambers can be unvented (closed) or vented (open), depending on whether or not they have some kind of pressure vent that allows passive air exchange with the atmosphere. Closed chambers prevent any air flow in or out, whereas open chambers allow free air flow and better mimic field conditions (Di Perta et al., 2020). Chamber methods, however, are known to influence environmental parameters such as radiation, evaporation, temperature, and wind speed, all of which impact the transport of $NH_3$ from the soil surface (Behera et al., 2013). As a result, these methods are typically used for qualitative rather than quantitative $NH_3$ measurements. Open chamber designs using absorbing sponges treated with acidic solutions, such as the design described by Wang et al. (2004), can continuously measure $NH_3$ emissions in the field. However, there remains a lack of validation or calibration for the absorbing sponge designs in quantifying $NH_3$ emissions under field conditions. Furthermore, the use of such sponges in chamber designs necessitates access to laboratory personnel capable of analyzing the $NH_3$ content in the collected samples.

The Dräger Tube Method (DTM) was developed as a simple and cost-effective alternative for quantifying $NH_3$ volatilization from soils, also covered by arable vegetation. It does not require a local power source or special laboratory equipment. This method allows measurements on smaller or heterogeneous fields (Pacholski et al., 2006; Roelcke et al., 2002). Daily $NH_3$ flux

is quantified by linearly interpolating between measurements at discrete point in time. The current DTM system consists of four conical stainless steel chambers connected by several short PTFE tubes. Ambient air is drawn from the chambers and passed through an $NH_3$ detector tube (Dräger Tube) from Drägerwerk AG & Co. KGaA (Lübeck, Germany) with the aid of a hand or automatic pump. The Dräger Tubes contain bromophenol blue, a pH indicator that turns blue as a result of the reaction with $NH_3$. The intensity of the blue coloration is proportional to the amount of reacting gas. Earlier comparisons of $NH_3$ fluxes measured in the laboratory and $^{15}N$ field studies showed good correlations, but the DTM underestimated the flux by an order of magnitude, which was attributed to the low air exchange rate (Rees et al., 1996; Roelcke et al., 2002). The DTM was later calibrated by Pacholski et al. (2006) with simultaneous measurements using the Integrated Horizontal Flux method (IHF). This calibration approach was later on validated by other comparative trials involving micrometeorological measurements (Gericke et al., 2011; Quakernack et al., 2012; Ni et al., 2015). Although a calibration was applied to the DTM, there remains considerable uncertainty surrounding its accuracy. Recent literature highlights an underestimation of $NH_3$ fluxes even after calibration, suggesting potential biases in the calibration process itself (Kamp et al., 2024). This indicates that the current calibration approach may not fully account for all factors influencing $NH_3$ emissions. An inherent underestimation of $NH_3$ fluxes by the DTM could mean that the DTM also has a high detection limit, which could lead to unmeasurable $NH_3$ mixing ratios in low-emitting plots. If this is the case, it makes sense to look for ways to improve the sensitivity of the DTM.

The DTM is susceptible to the same measurement errors that occur with other chamber systems for measuring $NH_3$. Wall effects caused by the adhesion of $NH_3$ to the chamber and tubing surface can lead to an underestimation or hysteresis of the mixing ratio measurements of up to 50 % (Sintermann et al., 2012). This is due to the fact that $NH_3$ is a highly reactive gas that can combine very quickly with other molecules. As a result, $NH_3$ is very soluble in water and adheres to even the smallest water film on any surface, which delays the path from the chamber system into the measuring device. This delay is greater at temperatures of 5 °C or less, and less at higher temperatures such as 25 °C, as the volatility of $NH_3$ increases at higher temperatures (Fogg, 1991). In the past, the DTM had been used to perform $NH_3$ concentration measurements with different materials and methods. Roelcke et al. (2002) and Richter (1972) originally used four tin chambers with a total surface area of 400 cm2 and inserted a polyethylene funnel into the chambers. Roelcke et al. (2002) used 35 cm PTFE tubing to connect the chambers and flushed 2–3 litres of air from the bottom surface through the chambers and into a used Dräger Tube each time before starting the measurements. This was intended for $NH_3$ to achieve a state of equilibrium in the chambers. This approach was further modified by Pacholski et al. (2006) by using stainless steel as the chamber material with a total surface area of 415 cm2 for the chambers. The rinsing volume to reach a state of equilibrium was set to 2 l in the latest version of his method (Pacholski, 2016). The measuring time ranges from 1–5 min. Wall effects of the chamber system, the short measurement duration and the low flow rate could all contribute to reduced measurement accuracy.

There have also been a number of studies aimed at improving or testing the detection sensitivity of different chamber measurement systems, but none of them directly testing the influence of different materials used for dynamic chambers on $NH_3$ mixing ratios (Di Perta et al., 2020). Yang et al. (2019) compared $NH_3$ measurements from four chamber methods with predicted values and found that the portable ammonia detector method had the highest detection sensitivity and the lowest detection limit of the four. Regarding the material used for the tubing that transports $NH_3$ inside the instrument, Shah et al.

(2006) tested the adsorption rate of $NH_3$ in five different tubing materials after 2 h at 1 and 10 parts per million (ppm) and at a flow rate of $10\,l\,min^{-1}$. They found no significant difference between the selected materials. The flow rate was much higher than that used in the DTM and the adsorption dynamics at lower and higher time intervals were not studied. A significant part of the contribution to lower capture efficiencies could also come from the use of a stainless steel surface of the chambers. In a tubing material experiment, Yokelson et al. (2003) observed a longer response time of $NH_3$ within their experiment when they replaced part of the PTFE tubing with a stainless steel tubing at room temperature. This delay increased further at a temperature of 5°C.

In addition to the chamber system, the Dräger Tubes themselves have an intrinsic standard deviation of 10-15 % for repeated measurements of the same $NH_3$ source (Drägerwerk AG, 2011). There are no publicly available results of tests on the sensitivity and detection accuracy of the Dräger Tubes. In addition, the original main purpose of the Dräger Tubes was the direct measurement of hazardous gas mixing ratio variations in the workplace or in enclosed spaces prior to entry, and the detection of gas leaks in process pipelines (Drägerwerk AG, 2011). For these applications, high accuracy and sensitivity are not required.

The main objective of this study was to test the detection accuracy, precision and sensitivity of the Dräger Tubes when used for the uncalibrated DTM. $NH_3$ was measured in various mixing ratios from 50 to 1500 parts per billion (ppb) with Dräger Tubes and a multicomponent gas analyzer based on quantum cascade laser spectroscopy (QCLS) (MGA[7], MIRO Analytical AG, Wallisellen, Switzerland). We chose a cavity ring-down spectrometer (CRDS) (G2103, Picarro, Inc., USA) as the reference device to compare the Dräger Tube and QCLS measurements to. The manufacturer of the CRDS used in this study does factory calibrations by using a so-called golden instrument as a reference standard, from which a specific calibration factor for each produced instrument is derived. Regular checks ensure the stability of the golden instrument's $NH_3$ calibration. The reliability of this calibration factor was independently confirmed by ab initio calculations using the HITRAN2012 database (Rothman et al., 2013) and by the national Physical Laboratory of the United Kingdom (Martin et al., 2016). The stability of the CRDS analyzers was tested in a large intercomparison experiment with 47 CRDS analyzers, and a typical drift of about 0.1 % slope per year was found (Yver Kwok et al., 2015). This high stability in measurements because of the low annual drift ensures that regular direct calibratrions in which the calibrations slope has to be changed is not necessary (Rella, 2017), and also makes the used CRDS from Picarro a suitable reference device to compare the other devices to.

The influence of tubing material and temperature on response time was tested under laboratory conditions with the QCLS. Due to the dependence of temperature on the adsorption of $NH_3$, it was expected that the heated tubes would perform better, i.e., feature a lower response time to changes in $NH_3$ mixing ratio. Finally, the material used for the chamber system was tested for effects on the measured $NH_3$ mixing ratios under field conditions. Uncalibrated DTM measurements with modifications to the chamber system were compared with measurements with the QCLS. The QCLS was able to display the $NH_3$ mixing ratios entering the system in real time. This minimized the risk of underestimating the $NH_3$ mixing ratios due to adsorption associated with short measurement times. The hypothesis was that the uncalibrated DTM would underestimate mixing ratios compared to the QCLS. The modifications tested on the Dräger system included changing the tubing material to polyester-polyurethane (PU) or Synflex 1300, wiping the inner surface of the chambers with ethanol after each use, and applying a thin PTFE film to the inner surface of the chambers.

## 2 Material and Methods

The study was divided into three experiments. The first experiment focused on quantifying the Dräger Tube detection accuracy, precision, and sensitivity for $NH_3$. The second experiment investigated the influence of tubing material and temperature on the response time. The third experiment evaluated the modifications to the chamber system during outdoor measurements.

### 2.1 Laboratory experimental setup

A sketch of the experimental setup can be found in Fig. 1. Compressed air free of $NH_3$ was humidified with a water bubbler to achieve ambient water vapor concentration. This air was mixed with $NH_3$ standard gas. The desired $NH_3$ and water vapor for the sample gas was achieved by regulating the flow of both gas tanks with two needle valves. The sample gas was led through a pump into a mass flow meter to set the flow rate. For the tubing material and the heating experiment, the tubing inserted between the regulated pump and the QCLS was replaceable. The fixed tubing was PTFE with an outer diameter of 6.35 mm. To eliminate the adsorption effects of the fixed tubing, sample gas was constantly flushed through the system. An excess port was installed after the replaceable tubing part to prevent overpressure in the gas analyzers. The tubing connected to the Dräger Tube acted as an additional excess port whenever the Dräger Tubes were not used. The pump behind the Dräger Tube was the Dräger X-act® 5000 Basic electric pump from Drägerwerk AG & Co. KGaA (Lübeck, Germany).

### 2.2 Quantification of the Dräger Tube detection accuracy, precision and sensitivity for $NH_3$

To determine the $NH_3$ detection accuracy of both the QCLS and the Dräger Tubes, humid air with a defined $NH_3$ mixing ratio was passed independently through both the QCLS and the CRDS in the laboratory setup. The mixing ratio was set at approximately 50, 100, 150, 200, 250, 300, 500, 1000, and 1500 ppb, respectively. The CRDS readings were used as a reference. The $NH_3$ readings of both the QCLS and the CRDS were allowed to stabilize before starting the Dräger Tube measurements. The Dräger Tubes were inserted into the $NH_3$-rich air excess port as shown in Fig. 1 and then air was pumped into the Dräger Tubes using the Dräger X-act 5000 Basic electric pump from Drägerwerk AG & Co. KGaA (Lübeck, Germany). A minimum of 10 and a maximum of 50 pump strokes were used for the measurements. Ten pump strokes were performed with a used Dräger Tube before each measurement. The Dräger Tube measurements were repeated three times for each mixing ratio level. The detection accuracy was determined by the difference in detected mixing ratios between the CRDS analyzer and the other instruments. The Dräger Tube measurements taken with more than 10 pump strokes (where 10 pump strokes equal 1.0 l of air volume) were scaled back to 1.0 l of air volume to make them comparable to the QCLS and CRDS measurements. A list of the instruments and materials used during the laboratory experiments can be found in Table A1 and A2.

### 2.3 Influence of tubing material and tubing temperature on $NH_3$ response time

The response time of the QCLS to $NH_3$ was tested using different tubing materials. We selected PTFE (CS - Chromatographie Service GmbH, Langerwehe, Germany), PU (Landefeld Druckluft und Hydraulik GmbH, Kassel-Industriepark, Germany) and Synflex 1300 (Megaflex Limited, Bideford, England) tubing for the tests. The tubing had an outer diameter of 6.35 mm and an

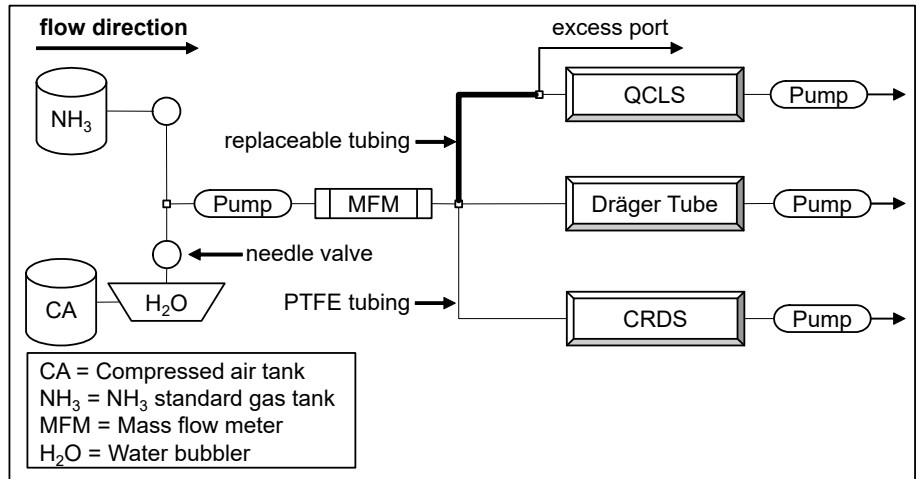

**Figure 1.** A sketch of the experimental setup used to test the NH$_3$ detection accuracy of the Dräger Tubes and QCLS and the performance of different tubing materials at room temperature and when heated. The tubing connected to the Dräger Tube acted as a second excess port whenever the Dräger Tubes were not used. The pump behind the Dräger Tube was the Dräger X-act® 5000 Basic electric pump.

inner diameter of 4.3 mm for PTFE and Synflex 1300 and an inner diameter of 4.2 mm for PU. The same laboratory setup was used as described in Section 2.2, but with a fixed NH$_3$ mixing ratio of 600 ppb. A 3 m segment of the respective tubing was connected between the excess port next to the QCLS and the mass flow meter (Fig. 1). The tubing performance was tested at

an ambient temperature of roughly 24 °C and then in a heated state at roughly 50 °C by wrapping a 5 m long 20 W aquarium heating cable (Dennerle Eco-Line ThermoTronic, Dennerle GmbH, Münchweiler, Germany) around the tubing and covering it with insulation material ArmaFlex AF-2-012 (inner diameter 12 mm, insulation thickness 13 mm). The response time was defined as the time required for the sensor to detect 10–90 % of total NH$_3$ at increasing mixing ratios, and the time required for the sensor to detect 90–10 % at decreasing mixing ratios.

**2.4    Outdoor experimental setup**

We tried different modifications of the chamber system for the outdoor experiment and compared the differences in the measured mixing ratios of the Dräger Tubes and the QCLS. The chamber system for DTM and QCLS was the same as that used in Pacholski et al. (2006) and the modification changes were applied to the chamber systems of both devices. In the first trial, we compared the NH$_3$ mixing ratios of both devices without any changes to the chamber system. In the second trial, we wiped the

inner surface of the chamber with 99 % ethanol before each measurement. Since NH$_3$ has a lower solubility in ethanol than in water, this was intended to replace the water film on the inner chamber surface. In the third trial, we replaced the PTFE tubing with PU tubing. Both PTFE and PU are hydrophobic. However, PU is much less expensive and more flexible, making it more practical to use during measurements. In the final trial, in an attempt to minimize potential water films on the inner surface

of the chamber, the inside of the chambers was covered with 0.05 mm thick PTFE sheet. The PTFE sheet was attached to the surface of the chamber with a double-sided adhesive tape.

Four boxes, with dimensions of 56.5 × 36.0 × 17 cm, containing agricultural soil were situated in close proximity to a laboratory building on the campus of Forschungszentrum Jülich, Germany. The location ensured that the QCLS had both power and shelter nearby in the event of rain. The soil used was agricultural soil (silty loam, pH 6.4) collected from arable land at the agricultural research site Klein-Altendorf near Bonn, Germany. The coordinates of the soil collection point were 50.61618° N, 6.99489° E. A ClimaVUE50 all-in-one weather sensor from Campbell Scientific (Logan, USA) and a CR300 data logger from Campbell Scientific (Logan, USA) were stationed near the soil boxes to record real-time weather data with a measurement interval of 1 min. Soil moisture was recorded with a TRIME PICO64 moisture sensor from IMKO Micromodultechnik GmbH (Ettlingen, Germany) connected to an HD2 mobile reader from IMKO Micromodultechnik GmbH (Ettlingen, Germany), and soil temperature was recorded with a digital thermometer (Checktemp® 1 HI98509, Hanna Instruments Woonsocket, RI, USA) prior to each $NH_3$ measurement. Four stainless steel soil rings with a diameter of 11.5 cm were placed on the soil of each box. A solid urea fertilizer with 46 % nitrogen content (Piagran 46®, SKW Stickstoffwerke Piesteritz GmbH, Lutherstadt Wittenberg, Germany) was used for the fertilized plots. Urea was applied only within the soil rings, and approximately 0.135 g was applied to each soil ring, corresponding to 60 kg N ha$^{-1}$. After fertilization, each soil box was evenly irrigated with 1–2 mm of water whenever the soil was determined to be too dry to dissolve the urea. Soil boxes were re-fertilized when $NH_3$ was no longer detectable by the Dräger Tubes. A list of the materials used during the outdoor experiment can be found in Table A3 and A4.

## 2.5 Outdoor measurements

The QCLS and its chamber system were connected to an electric pump and flow meter to maintain an air exchange rate of approximately $1.0 \, l \, min^{-1}$. The Dräger X-act® 5000 Basic pump used for the DTM takes approximately 1 min for 10 strokes of 0.1.0 l of air each stroke, which also corresponds to an air exchange rate of approximately $1.0 \, l \, min^{-1}$. Prior to the start of each measurement with the QCLS, the chamber system was placed on the soil rings of the respective box and flushed with air for 30 min. After a further 30 min, the indicated mixing ratio was recorded. At the same time, a second identical chamber system was flushed with another electric pump on another set of soil rings in preparation for the next measurement. While we waited a total of 60 minutes during each measurement cycle, we only recorded the last few minutes of the QCLS readings. This ensured that the system reached a steady state before data collection began. DTM measurements were taken immediately after the completion of each QCLS measurement using the same chamber system. Measurements were performed according to the instructions from Pacholski (2016).

The air volume passing through the Dräger Tube depends on the number of pump strokes performed and was therefore always converted back to 10 pump strokes (equivalent to 1.0 l of air) for comparability.

## 2.6   Data Analysis

Data transformation and statistical analysis were performed using R version 4.3.2. In the intercomparison test of all instruments for $NH_3$ mixing ratios, the detection accuracy ($y_i$), where $i$ is either Dräger Tubes or QCLS, was defined as:

$$y_i = \frac{NH_{3_i} - NH_{3_{CRDS}}}{NH_{3_{CRDS}}} \cdot 100 + 100 \tag{1}$$

Detection precision was defined as the relative standard deviation of the measurements. Detection sensitivity was defined as the beta coefficient of a linear regression fitted to predict measured CRDS $NH_3$ mixing ratios from measured $NH_3$ mixing ratios of either the QCLS or Dräger Tubes.

A modified Hill function with an offset (Hill1, see Eq. 2) was fitted to the $NH_3$ response curves of the response time tests using OriginPro 2022b (64-bit) SR1 version 9.9.5.171. Where $x$ was the duration in seconds; $y$ was the $NH_3$ mixing ratio at a given duration; START was the $NH_3$ mixing ratio at $x0$; END was the maximum $NH_3$ mixing ratio for rising response curves or the minimum for falling response curves; $k$ was the duration to reach 50 % of maximum $NH_3$ mixing ratios; $n$ was the Hill coefficient. Response time was defined as the duration from $y_{10\%}$ ($EC_{10}$) to $y_{90\%}$ ($EC_{90}$) for rising curves and from $EC_{90}$ to $EC_{10}$ for falling curves (see Eq. 3 and Eq. 4). To test the response time of rising response curves for statistical significance, a one-way ANOVA was used. For falling response curves, which did not follow a normal distribution, the Kruskal–Wallis rank sum test was used.

$$y = START + (END - START) \cdot \frac{x^n}{k^n + x^n} \tag{2}$$

$$EC_{10} = \frac{k}{9^{\frac{1}{n}}} \tag{3}$$

$$EC_{90} = k \cdot 9^{\frac{1}{n}} \tag{4}$$

Finally, linear regression was used to compare the differences between the DTM and QCLS $NH_3$ measurements for each outdoor modification trial.

## 3   Results

### 3.1   Quantification of the Dräger Tube detection accuracy, precision and sensitivity for $NH_3$

The detection accuracy and precision of the Dräger Tubes and QCLS measurements are displayed in Fig. 2. The QCLS measured slightly higher $NH_3$ mixing ratios compared to the CRDS and Dräger Tubes and had a detection accuracy of 97–114 % across the different $NH_3$ levels. The Dräger Tubes on the other hand measured lower mixing ratios and had a detection accuracy of 0–72 % across the different $NH_3$ levels. The detection accuracy was higher at high $NH_3$ levels and decreased to 0

**Table 1.** Relevant statistical information for the detection accuracy and precision of the Dräger Tubes and QCLS measurements. df = degrees of freedom.

|  | QCLS | Dräger Tubes |
|---|---|---|
| Detection Accuracy (%) | 97–114 | 0–72 |
| Detection Precision (%) | 0.02–1.80 | 0–115.47 |
| Beta Coefficient | 1.12 | 0.72 |
| 95% Confidence Interval | [1.11, 1.13] | [0.69, 0.76] |
| t-value (df = 34) | 372.11 | 41.03 |
| p-value (t-test) | < 0.001 | < 0.001 |
| $R^2$ | 1.00 | 0.98 |
| F-value (df = 1, 34) | $1.38 \times 10^5$ | 1683.44 |
| p-value (F-test) | < 0.001 | < 0.001 |
| Adjusted $R^2$ | 1.00 | 0.98 |
| Shapiro-Wilk W | 0.75 | 0.72 |
| p-value (Shapiro-Wilk) | < 0.001 | < 0.001 |

235 % at lower levels. The detection precision ranged between 0.02–1.80 % for the QCLS measurements and between 0–115.47 % for the Dräger Tubes (Table 1).

The mean $NH_3$ detection sensitivity determined a beta coefficient of a fitted linear model to predict measured CRDS $NH_3$ mixing ratios with measured $NH_3$ mixing ratios using the QCLS or Dräger Tubes, which was statistically significant and 1.12 for the QCLS and 0.72 for the Dräger Tubes. However, the data of both the QCLS and Dräger Tube mixing ratios did not 240 follow a normal distribution.

### 3.2 Influence of tubing material and tubing temperature on $NH_3$ response time

The main effect of the tubing material and temperature for rising $NH_3$ levels (Fig. 3a) was significant and large according to the performed one-way ANOVA (Table 2). The response time of heated PTFE tubing was 10.71 min (standard deviation (SD) 0.92) and significantly shorter than both unheated and heated PU and Synflex tubing, but not significantly shorter than 245 unheated PTFE tubing. The response time of unheated Synflex tubing was 51.25 min (SD 8.25) and significantly longer than both heated and unheated PTFE tubing and heated PU tubing. The main effect was significant for falling $NH_3$ levels (Fig. 3b) according to the performed Kruskal–Wallis test (Table 2). The response time of heated PTFE tubing was 6.39 min (SD 0.23) and significantly shorter than both unheated and heated Synflex tubing, but not significantly shorter than the other tubing materials. The response time of unheated Synflex tubing was 25.11 min (SD 0.81) and significantly longer than unheated PU 250 and heated PTFE tubing.

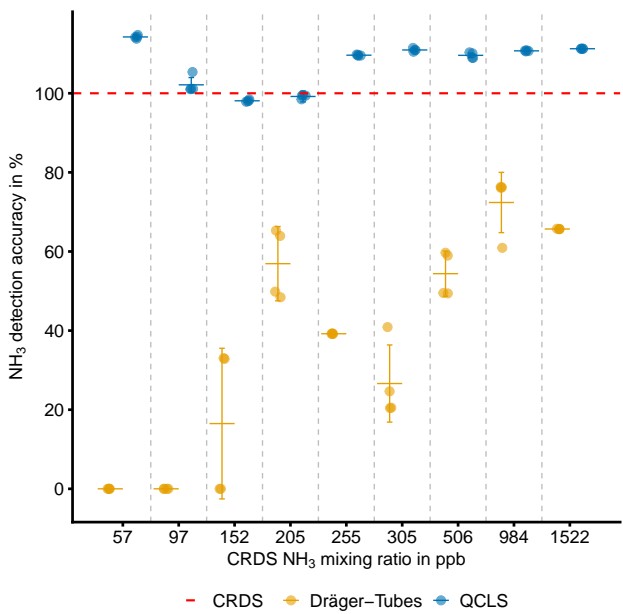

**Figure 2.** Detection accuracy of $NH_3$ with the Dräger Tube and QCLS relative to the $NH_3$ mixing ratio measurement of the CRDS. Error bars indicate the detection precision as one standard deviation.

**Table 2.** Relevant statistical information for the effect of tubing material and temperature on rising and falling $NH_3$ levels. df = degrees of freedom.

|  | Rising $NH_3$ Levels | Falling $NH_3$ Levels |
|---|---|---|
| Main Effect Test | One-way ANOVA | Kruskal–Wallis |
| F-value (df = 5, 13) | 12.97 | 332.74 |
| p-value | $< 0.001$ | $< 0.001$ |
| Eta$^2$ | 0.83 | 0.89 |
| 95% Confidence Interval | [0.61, 1.00] | [0.71, 0.96] |

## 3.3 Chamber system modifications during outdoor measurements

A linear model was fitted to predict DTM $NH_3$ measurements from QCLS $NH_3$ measurements during four different trials where the chamber system was left unchanged or was slightly modified (Fig. 4). In the unchanged chamber system trial, the model explained a statistical significant and substantial proportion of variance. The model's intercept, corresponding to QCLS = 0 ppm, was at -0.09 ppm. The unchanged chamber system had the highest beta coefficient of the four trials 3.

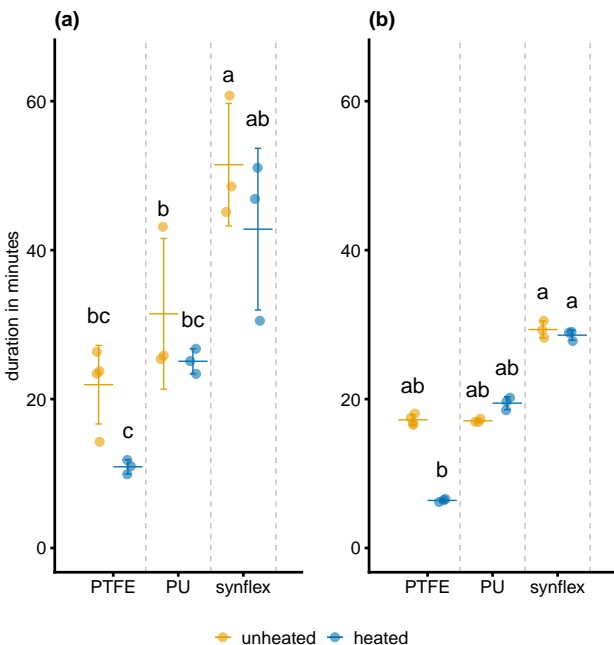

**Figure 3.** Response time for NH$_3$ mixing ratios measured with the QCLS for (a) increasing from 10–90 % and (b) decreasing from 90–10 % of the NH$_3$ target level, using different tubing materials unheated at ambient temperature or heated with a wire to 50 °C. Letters above error bars indicate significant differences between groups, as determined by one-way ANOVA (a) and Kruskal-Wallis test (b). Groups sharing the same letter are not significantly different. Error bars indicate the standard deviation.

During the trial where the inner chamber surfaces were cleaned with ethanol, the model explained a statistically significant, but only moderate proportion of the variance. The model's intercept was at 0.32 ppm. This trial had the second lowest beta coefficient of the four trials.

During the trial where the PTFE tubing was replaced with PU tubing, the model explained a statistically significant and substantial proportion of variance. The model's intercept was at 0.36 ppm. This trial had the second highest beta coefficient of the four trials.

During the trial where a PTFE film was attached to the inner chamber surfaces, the model explained a statistical significant and substantial proportion of variance. The model's intercept was at 0.19 ppm. This trial had the lowest beta coefficient of the four trials.

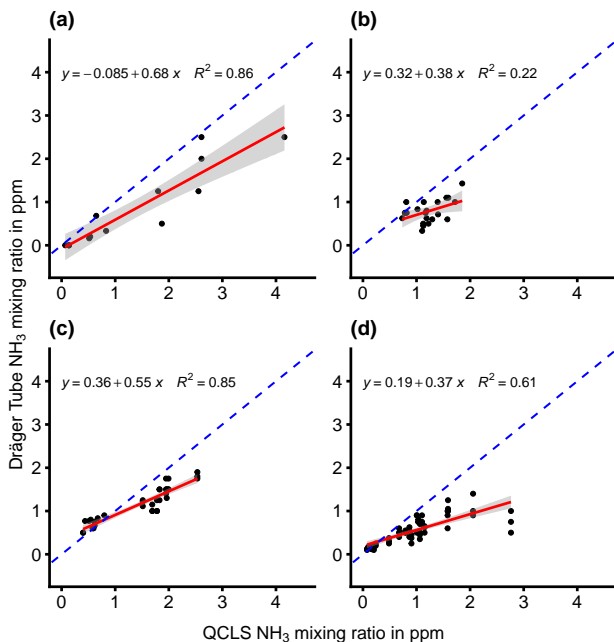

**Figure 4.** Linear regression relationships between Dräger Tube and MIRO measurements of the $NH_3$ mixing ratio in ppb under different measurement conditions. (a) = unchanged DTM chambers connected to PTFE tubing; (b) = wiping of chambers with ethanol; (c) = connected to PU tubing and (d) = chambers with PTFE film attached to the inner surface.

**Table 3.** Relevant statistical information for the chamber system modifications during outdoor measurements.

|  | unchanged chambers | wiped with ethanol | PU tubing | PTFE film attached |
|---|---|---|---|---|
| F-value of $R^2$ | 72.38 (df = 1,12) | 5.52 (df = 1,20) | 182.02 (df = 1, 33) | 94.13 (df = 1,61) |
| p-value of $R^2$ | $< 0.001$ | 0.029 | $< 0.001$ | $< 0.001$ |
| 95% confidence interval of intercept | [-0.4, 0.23] | [-0.10, 0.74] | [0.23, 0.50] | [0.11, 0.28] |
| p-value of intercept | 0.564 | 0.123 | $< 0.001$ | $< 0.001$ |
| beta coefficient | 0.68 | 0.38 | 0.55 | 0.37 |
| 95% confidence interval of beta | [0.50, 0.85] | [0.04, 0.71] | [0.46, 0.63] | [0.29, 0.44] |
| t-value of beta | 8.51 | 2.35 | 13.49 | 9.70 |
| p-value of beta | $< 0.001$ | 0.029 | $< 0.001$ | $< 0.001$ |

## 265    4   Discussion

### 4.1   Quantification of the Dräger Tube detection accuracy, precision and sensitivity for $NH_3$

Slightly higher $NH_3$ mixing ratios were measured with the QCLS than with the CRDS. The detection accuracy was between 97 % and 114 % for the various $NH_3$ values. The mean detection sensitivity was 0.89. This means that the QCLS slightly

overestimated the $NH_3$ mixing ratios in most cases compared to the CRDS. This is in line with the results of the case study by the manufacturer of the QCLS (MIRO Analytical, 2021), which also compared $NH_3$ measurements of the MGA with a cavity ring down spectroscopy device and found an average detection sensitivity of 0.748 at mixing ratios between 3 and 9 ppb. Since the detection accuracy was good even at low $NH_3$ mixing ratios and comparable to that of the CRDS, it was concluded that the QCLS could be safely used for response time and field measuring.

A significant drop in detection accuracy in Dräger Tubes was observed between 205 and 305 ppb. Measurements at 97, 152, and 205 ppb were conducted later in March 2023, while the other measurements were performed in October 2022. The laboratory temperature in March was 5 °C lower, which may have influenced the results. Additionally, differences in the batches of Dräger Tubes used could have affected the measurement quality. However, the most likely explanation is observer error. Dräger Tubes lack the sensitivity to detect changes within a range of ±100 ppb. $NH_3$ concentrations are determined by measuring the distance of discoloration on the detector tubes, leaving the precise endpoint of the discoloration subject to observer judgment. In some cases, the discoloration was only a very light blue, further complicating accurate observations.

The Dräger Tubes had a detection accuracy of 0 % at lower $NH_3$ mixing ratios and a detection accuracy of up to 72 % at higher mixing ratios. This means that the Dräger Tubes considerably and significantly underestimated the $NH_3$ mixing ratios compared to the CRDS device, with a trend of increasing underestimation at decreasing $NH_3$ levels. It was already known that the uncalibrated DTM underestimates $NH_3$ and is not suitable for quantitative measurements of $NH_3$, but previous work on the DTM has cited the low air flow rate of the pump as the most likely reason for the underestimation of $NH_3$ mixing ratios (Pacholski et al., 2006; Roelcke et al., 2002). Svensson and Ferm (1993) also found a direct relationship between $NH_3$ concentrations in their chamber and the air flow rate. To address this limitation, Pacholski et al. (2006) proposed a calibration approach that accounts for wind speed, which could potentially make the DTM suitable for quantitative measurements. However, the flow rate of the pumps in this test was set to that of the Dräger pump, which was approximately $1.0\,l\,min^{-1}$, for all instruments and $NH_3$ was constantly flowing at a uniform rate throughout the entire tubing line, so there must be other factors contributing to the lower detection accuracy of the Dräger Tubes compared to the other devices.

One reason for this could be the short measuring time of the Dräger Tubes. While we gave the other devices sufficient time to measure 100 % of the $NH_3$ target value, this was not possible with the Dräger Tubes, as the tubes would be saturated with $NH_3$ if air was pumped through the tubes for too long. Another reason could be that the Dräger Tubes were not designed for precise atmospheric measurements. According to the official instructions, only 10 strokes are intended for the measurement with the Dräger Tubes, which corresponds to an air volume of 1.0 l. Throughout the experiment, we used $NH_3$ detector tubes whose indicator line ranged from 250–3000 ppb (Dräger-Tube: Ammonia 0.25/a, Tab. A1). At values below 300 ppb, therefore, more than the recommended amount of 1.0 l of air would have to be pumped through the Dräger Tube for $NH_3$ to be detectable. This would explain the sharp drop in detection accuracy in the Dräger Tubes for mixing ratios below 200 ppb, where 5 l of air had to be pumped through the Dräger Tube instead of 1.0 l (Fig. 2). The original use of the Dräger Tubes was to measure excess $NH_3$ concentrations after fumigation of enclosed spaces, such as containers (Drägerwerk AG, 2011). The Dräger Tubes were therefore not designed for high detection accuracy, especially at lower $NH_3$ concentrations.

The originally assumed detection limit of the Dräger Tubes, as used in the DTM, was 50 ppb. However, the results suggest that the detection limit is instead somewhere between 152 ppb and 205 ppb. The originally proposed detection limit was

based on the assumption that increasing the number of pump strokes per measurement to 50 instead of 10 would proportionally improve the detection limit of the Dräger Tubes, which indicated 250 ppb on the lowest indicator line, down to 50 ppb (Roelcke, 1994). Since the previous detection limit was based on an assumption and not on empirical measurements, we suggest that a detection limit between 152 and 205 ppb is more correct.

A higher detection limit than originally assumed could lead to erroneous $NH_3$ flux measurements from the calibrated DTM

in two different ways. The first is that the actual $NH_3$ mixing ratios at the fertilized soil surface are below the detection limit and are no longer detectable. This case would cause a higher flux detection limit and the $NH_3$ fluxes would be underestimated. The second case is that the mixing ratios at the fertilized soil surface are detectable and the background mixing ratios are unusually high, but still below the detection limit. In this case, the $NH_3$ fluxes would be overestimated. Therefore, instruments with high measurement sensitivity are preferred.

To calculate the flux detection limit of the calibrated DTM from the mixing ratio detection limit, we chose to apply the above-mentioned calibration equation (Eq. 6) from Pacholski et al. (2006), which was recently identified by Kamp et al. (2024) to have considerable uncertainty and lead to an underestimation of $NH_3$ fluxes. Since $NH_3$ fluxes derived from this equation are strongly influenced by wind speed during the measurement period, the flux detection limit also varies significantly depending on the wind conditions at the time of measurement. Table 4 shows an estimate of the effect that a detection limit of 175 ppb

can have on the daily $NH_3$ fluxes measured and calculated with the calibrated DTM. Typically, DTM measurements should be taken multiple times throughout the day, and the fluxes are then interpolated between measurement points to account for diurnal variation. However, for our estimates, we assumed constant $NH_3$ concentrations over the course of the day. $NH_3$ mixing ratios above the soil surface were assumed to be near the detection limit of 175 ppb, with constant wind speeds maintained throughout the day. This number was divided by the $\beta$-coefficient of 1.36 from the detection sensitivity experiment and converted to an

observed mixing ratio of 126 ppb. Average wind speed levels were set starting at 0.1 m s$^{-1}$ and then matching the upper limits of the Beaufort scale, ranging from calm to gentle breeze. They were capped at 4 m s$^{-1}$, since that is the upper limit for which the DTM was calibrated for (Pacholski et al., 2006). In addition, the temperature was kept constant at 20 °C and the air pressure at 1013 hPa. The calibrated DTM flux rate $F_{DTM}$ (in kg N ha$^{-1}$) was calculated according to Pacholski et al. (2006) as follows:

$$F_{DTM} = \text{volume} \cdot |\text{conc.}| \cdot 10^{-9} \cdot \rho_{NH_3} \cdot U_N \cdot U_F \cdot U_Z \tag{5}$$

Where volume is the air volume sucked through the chambers, in this case 5 l; $|\text{conc.}| \cdot 10^{-9}$ is the mixing ratio of $NH_3$ in ppb as was displayed in the Dräger Tubes, in this case 630 ppb for 5 l; $\rho_{NH_3}$ is the temperature-dependent density of $NH_3$ at 20 °C in kg l$^{-1}$; $U_N$ is the molecular weight conversion factor of $NH_3$ to N; $U_F$ is the surface area conversion factor from the chamber surface area of 415 cm$^{-2}$ to ha; $U_Z$ is the time conversion factor from seconds to days. Finally, the calibrated flux rate $F_{cal}$ was

calculated by incorporating wind speed $\nu$ according to the following equation:

**Table 4.** Possible underestimation of daily NH$_3$ fluxes assuming constant wind speeds over the course of the day and if the NH$_3$ mixing ratio at the ground surface is 175 ppb, making it undetectable by the calibrated DTM, and overestimation assuming background mixing ratios are also 175 ppb and also undetectable.

| Wind speed in m s$^{-1}$ | Beaufort scale | Wind description | Detection limit of daily NH$_3$ fluxes in kg N ha$^{-1}$ d$^{-1}$ |
|---|---|---|---|
| 0.1 | 0 | Calm | 0.05 |
| 0.2 | 0 | Calm | 0.08 |
| 1.5 | 1 | Light air | 0.25 |
| 3.3 | 2 | Light breeze | 0.40 |
| 4 | 3 | Gentle breeze | 0.44 |

$$F_{cal} = \exp\left(0.444 \cdot \ln\left(F_{DTM}\right) + 0.59 \cdot \ln\left(\nu\right)\right) \tag{6}$$

It can be seen that a detection limit of 175 ppb could lead to a detection limit of daily NH$_3$ fluxes of 0.05–0.44 kg N ha$^{-1}$ d$^{-1}$. If these numbers were applied to an emission factor of 15 % for urea (Asman, 1992) and an application rate of 60 kg N ha$^{-1}$, it would lead to a daily flux error of between 0.06 and 4.8 % in relation to the application rate. If, on the other hand, these figures were applied to a fertilizer with a low NH$_3$ emission factor, such as calcium ammonium sulphate with an emission factor of 2 % (Asman, 1992), this would lead to a daily flux error of between of between 6 and 36.7 % in relation to an application rate of 60 kg N ha$^{-1}$. Pacholski et al. (2006) previously determined a mean relative error of 17 $\pm$ 5 % for NH$_3$ losses using the calibrated DTM. Undetectable NH$_3$ fluxes due to the less sensitive detection limit would however result in a higher mean relative error than previously assumed. Daily underestimation of NH$_3$ fluxes while using fertilizer with an emission factor of 15 % and average wind speeds of 1.5 m s$^{-1}$ would exceed the mean relative error of 17 $\pm$ 5 % for NH$_3$ losses after six days. Daily underestimation of NH$_3$ fluxes while using fertilizer with an emission factor of 2 % on the other hand and average wind speeds of 1.5 m s$^{-1}$ would exceed the mean relative error of 17 $\pm$ 5 % for NH$_3$ losses after just one day. The use of the calibrated DTM should therefore only be considered if other measurement alternatives are not feasible and if high NH$_3$ fluxes are to be expected during the entire measurement campaign. Therefore, when combined with passive samplers, it was recommended to use the calibrated DTM with a high emission source for measurements of absolute emissions (Pacholski, 2016). In a recent comparative study by Kamp et al. (2024), the calibrated DTM also underestimated emissions compared to micrometeorological measurements while wind tunnel measurements tended to depend on the air exchange rate. Between micrometeorological methods final emission varied by 30 %. The error evaluation in Table 4 assumes the calibration is accurate and unbiased. To fully validate this approach, the calibration itself must also be tested. Since the calibration heavily depends on accurate wind speed measurements, any inaccuracies could introduce additional bias to the NH$_3$ readings. Moreover, the calibration equation may not be universally applicable to varying field conditions, including differences in soil types, fertilizer

application methods, or environmental factors. Consequently, additional comparative measurements are required for a more comprehensive and conclusive assessment of the calibrated DTM.

In addition to the inherent measurement errors associated with the Dräger Tubes themselves, field implementation of the method introduces further sources of potential error. One significant limitation is the relatively small surface area of the chambers, which prevents the DTM from accounting for soil heterogeneity when estimating emissions over larger plots. Furthermore, the application of slurry or granular fertilizers may not be uniform across the plot, adding to variability. Another concern is the risk of missing diurnal variability in fluxes if sampling intervals are too infrequent. Pacholski (2016) recommends placing chambers on soil rings to minimize soil disturbance and to ensure precise application of fertilizers within the measured area. The area within the soil rings can be covered with a lid during fertilizer application and then manually fertilized with a precise amount. Measurements should be taken five times a day: early morning shortly after sunrise, late morning, early afternoon, late afternoon, and shortly before sunset. Whether larger chambers, a greater number of soil rings distributed across the plot, or more frequent measurements would enhance the accuracy of the DTM in field conditions remains uncertain and requires further investigation.

## 4.2 Influence of tubing material and tubing temperature on $NH_3$ response time

The response time was longest in both heated and unheated Synflex tubing for both increasing and decreasing $NH_3$ mixing ratios. The response time was shortest in heated PTFE tubing, although we could not find a significant reduction in response time between heated and unheated tubing and between PTFE and PU. Whitehead et al. (2008) on the other hand was able to find a reduction in response time for heated PTFE tubing. However, they measured the response time at lower $NH_3$ mixing ratios, a much shorter measurement interval of 300 seconds and with a Quantum Cascade Laser Absorption Spectrometer, which has a resolution of up to 10 Hz, which might together contribute to the differences in results. The lower sample size of 3 might have also caused a possible false negative in the results. It was also difficult to keep $NH_3$ levels constant between comparisons, but they did not differ by more than 20 %. On the other hand, Shah et al. (2006) could not find a significant difference in $NH_3$ adsorption to other plastic tubing materials at air flow rates of $10 \, \mathrm{l \, min^{-1}}$ either.

## 4.3 Chamber system modifications during outdoor measurements

In both the unchanged and PU tubing chamber system, the linear model of the DTM was able to predict QCLS measurements with the highest coefficient of determination, followed by PTFE coated chamber, and finally by chambers wiped with ethanol. The unchanged chamber system also had the highest beta coefficient out of the four trials, closely followed by PU tubing, then PTFE coating, and finally wiping with ethanol. Continuing to use unchanged chamber systems is therefore the best choice out of the four options. While PU tubing performed similarly to PTFE tubing and is less expensive, PTFE tubing is well known for its low water absorption and low permeability to gases and moisture vapor (Harper, 2000). The advantage of the lower cost of PU is negligible in the case of the DTM because the total length of tubing does not exceed 3 m. However, it is recommended that the PTFE tubing be replaced periodically because degradation over time is known to increase response time and increase losses of gases such as $H_2O$ (Lee et al., 1991; Whitehead et al., 2008).

Wiping the inner surface of the chamber with ethanol reduced the performance of the chamber system. Dry wiping of the chamber surfaces is therefore preferred. The lower performance compared to the QCLS is likely due to the fact that the ethanol did not completely evaporate from the surface during the Dräger Tube measurements, while the ethanol from the chamber system used for the QCLS completely or mostly evaporated during the 30 min measuring period. The use of a PTFE sheet for the inner chamber surface also decreased the detection sensitivity. A halocarbon wax coating could be used instead of a PTFE

film for future testing, as it was found that a halocarbon wax coating of a stainless steel surface was able to improve the travel time of $NH_3$ to a similar level as PTFE, even at lower temperatures (Yokelson et al., 2003). It is also worth exploring whether active passivation of the chamber surface and inner tubing surface with 1H,1H-perfluorooctylamine could similarly enhance the sensitivity of Dräger Tubes, since Roscioli et al. (2016) discovered a reduction in response time from 30 s to 2 s of their Dual Quantum Cascade Laser instrument for 90 % $NH_3$ recovery.

**4.4   Potential future options for quantifying $NH_3$ emissions in heterogeneous, small plots without power supply**

    A similar method to the calibrated DTM that could be considered for the quantification of $NH_3$ in heterogeneous, small plots without power supply is the dositube method (van Andel et al., 2017). It uses an $NH_3$ detector tube similar to the calibrated DTM, but instead places the tube directly into a semi-open chamber and allows it to passively absorb the $NH_3$ over a longer period of time. The advantages over the calibrated DTM would be less manpower and a longer time-weighted average of $NH_3$

loss. This would allow detection of lower fluxes. The dositube method showed good agreement in $NH_3$ loss estimates when compared to wind tunnel measurements but has not yet been validated with a micrometeorological or mass balance method. The use of diffusive passive samplers, such as ALPHA samplers from the UK Centre for Ecology & Hydrology, in combination with a backward Lagrangian stochastic (bLs) model, presents a promising option for future $NH_3$ flux measurements, provided there is experienced laboratory staff. Carozzi et al. (2013) reported that the uncertainty associated with ALPHA samplers combined

with bLs was comparable to other direct flux measurement techniques. Furthermore, Pedersen et al. (2018) demonstrated that emission fluxes derived from ALPHA samplers and bLs were consistent with those obtained through classical mass-balance measurement methods.

**5   Conclusions**

    This paper evaluated the detection accuracy, precision, and sensitivity of the uncalibrated DTM $NH_3$ measurements and ex-

415 plored potential chamber system improvements. It was found that the Dräger Tubes used for the uncalibrated DTM were underestimating the measured concentrations, had decreasing detection accuracy at lower mixing ratios, and had higher detection limits in the range of 152-205 ppb than initially assumed. This conversely also has an influence on the $NH_3$ flux detection limit of the calibrated DTM. The calibrated DTM is therefore unsuitable for measurements on low $NH_3$ emitting (acidic) soils, under low temperature conditions, with low $NH_3$ emitting N fertilizers such as calcium ammonium nitrate or fertilizers

combined with inhibitors, and for experiments with low N application rates. However, the use of the calibrated DTM could still be considered in field experiments where high emissions are expected and other more reliable alternatives are not feasible.

**Table A1.** $NH_3$ measuring devices used during the experiments.

| Device | Company | Measurement technique |
|---|---|---|
| Dräger-Tube: | **Drägerwerk AG & Co. KGaA** | bromophenol blue |
| Ammonia 0.25/a | Moislinger Allee 53-55 | pH-indicator |
| | 23558 Lübeck Germany | |
| MGA[7] | **MIRO Analytical AG** | direct laser absorption |
| | Widenholzstrasse 1 | spectroscopy |
| | CH-8304 Wallisellen | |
| | Switzerland | |
| G2103 | **Picarro, Inc.** | cavity ring-down |
| | 3105 Patrick Henry Dr. | spectroscopy |
| | Santa Clara, CA 95054 | |
| | USA | |

Unfortunately, there are no feasible alternatives for small plot $NH_3$ measurements yet. The development of similar chamber or easy-to-use measurement methods that are inexpensive, mobile, and have a low detection limit is therefore encouraged. Methods like the dositube approach and the ALPHA sampler combined with the bLs model show promise as future options for $NH_3$ flux measurement. However, they require further validation through experiments under diverse field conditions and in comparison with alternative measurement techniques. Detection accuracy, precision, and sensitivity should be compared with high-precision real-time measurement techniques such as cavity ring-down spectroscopy or direct laser absorption spectroscopy. Further assessment of the DTM in comparison with a reference method involving also the calibration approach is desirable for a comprehensive and conclusive evaluation of this measurement approach.

This study also identified options that should be excluded or used in the development of a new chamber method. Dry wiping of the chambers should be preferred over the use of ethanol. The use of an external heating source in combination with PTFE could improve response times for $NH_3$ measurements and could be implemented for outdoor use of open dynamic chamber systems. However, the additional use of heating wires around the tubing would require careful preparation of the tubing and an additional portable power supply, which would complicate handling and limit the mobility of the chamber system.

*Data availability.* All the data and R scripts used in this work can be accessed from: https://doi.org/10.26165/JUELICH-DATA/0LAIFH

**Appendix A**

**Table A2.** Tubing material used during the experiments.

| Tubing material | Company | Product number | Diameter |
|---|---|---|---|
| PTFE | **CS - Chromatographie Service GmbH** Am Parir 27 (Gewerbegebiet) 52379 Langerwehe Germany | 198026-01 | outer diameter: 6.35 mm inner diameter: 4.3 mm |
| Polyester-polyurethane | **Landefeld Druckluft und Hydraulik GmbH** Konrad-Zuse-Straße 1 34123 Kassel-Industriepark Germany | PUN 1/4 SCHWARZ | outer diameter: 6.35 mm inner diameter: 4.2 mm |
| Synflex 1300 | **Megaflex Limited** Old Rectory Landcross Bideford Devon EX39 5JA United Kingdom | DEK1/4 | outer diameter: 6.35 mm inner diameter: 4.3 mm |
| PTFE | **Landefeld Druckluft und Hydraulik GmbH** Konrad-Zuse-Straße 1 34123 Kassel-Industriepark Germany | TFL 8X6 NATUR | outer diameter: 8 mm inner diameter: 6 mm |
| Polyester-polyurethane | **Landefeld Druckluft und Hydraulik GmbH** Konrad-Zuse-Straße 1 34123 Kassel-Industriepark Germany | PU 8X6 NATUR | outer diameter: 8 mm inner diameter: 6 mm |

**Table A3.** Material used for the unmodified chamber system.

| Device/Material | Company | Product number | Specs |
|---|---|---|---|
| PTFE tubing | **Landefeld Druckluft und Hydraulik GmbH** Konrad-Zuse-Straße 1 34123 Kassel-Industriepark Germany | TFL 8x6 NATUR | outer diameter: 8 mm diameter: 6 mm |
| Stainless steel chambers | **Metallindustriewerk Heinr. Hofmann GmbH** Seekoppelweg 6 24113 Kiel Germany | "V2A-Ringe 1.5 mm stark" | custom made |
| Stainless steel soil rings | **Metallindustriewerk Heinr. Hofmann GmbH** Seekoppelweg 6 24113 Kiel Germany | "V2A-Kammersystem pieces)" | custom made |
| Y push in fitting | **Landefeld Druckluft und Hydraulik GmbH** Konrad-Zuse-Straße 1 34123 Kassel-Industriepark Germany | IQSY 80 | 8 mm diameter: 6 mm |
| Hose clamps | **Carl Roth GmbH + Co. KG** Schoemperlenstr. 3-5 76185 Karlsruhe Germany | AEH1.1 | outer diameter: 8–12 mm |
| Dräger X-act® 5000 Basic | **Drägerwerk AG & Co. KGaA** Moislinger Allee 53-55 23558 Lübeck Germany | Dräger X-act® 5000 Basic | |
| Dräger Accuro hand pump | **Drägerwerk AG & Co. KGaA** Moislinger Allee 53-55 23558 Lübeck Germany | Dräger Accuro | |
| Deluxe tube opener Dräger | **Drägerwerk AG & Co. KGaA** Moislinger Allee 53-55 23558 Lübeck Germany | Dräger TO 7000 | |

**Table A4.** Other material and devices used.

| Device/Material | Company | Product number | Specs |
|---|---|---|---|
| Checktemp® 1 digital thermometer | **Hanna Instruments Deutschland GmbH** An der Alten Ziegelei 7 89269 Vöhringen Germany | HI98509 | |
| Soil moisture sensor | **IMKO Micromodultechnik GmbH** Am Reutgraben 2 D-76275 Ettlingen Germany | TRIME PICO 64 | |
| Soil moisture logger | **IMKO Micromodultechnik GmbH** Am Reutgraben 2 D-76275 Ettlingen Germany | HD2 | |
| All in one weather station | **Campbell Scientific Ltd.** Fahrenheitstraße 13 28359 Bremen Germany | ClimaVUE50 | |
| Weather data logger | **Campbell Scientific Ltd.** Fahrenheitstraße 13 28359 Bremen Germany | CR300 | |
| MIRO Field Enclosure | **MIRO Analytical AG** Widenholzstrasse 1 CH-8304 Wallisellen Switzerland | | |
| Heating wire | **Dennerle GmbH** Industriestr. 4 66981 Münchweiler/Rodalb Germany | Dennerle Eco-Line ThermoTronic | 20 W |
| ArmaFlex | **Armacell GmbH** Robert-Bosch-Straße 10 48153 Münster Germany | ArmaFlex AF-2-012 | inner diameter: 12 mm insulation thickness: 13 mm |
| Virginal PTFE Sheet | **Hightechflon GmbH & Co. KG** Macairestr. 4 78467 Konstanz Germany | PTFE.SFL.005.VIR | thickness: 0.05 mm |

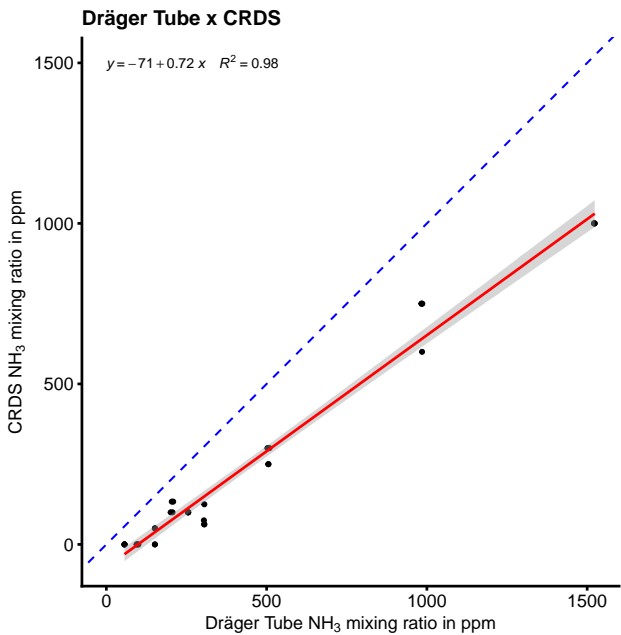

**Figure A1.** Linear regression plot between Dräger Tube and CRDS NH₃ measurements for chapter 3.1

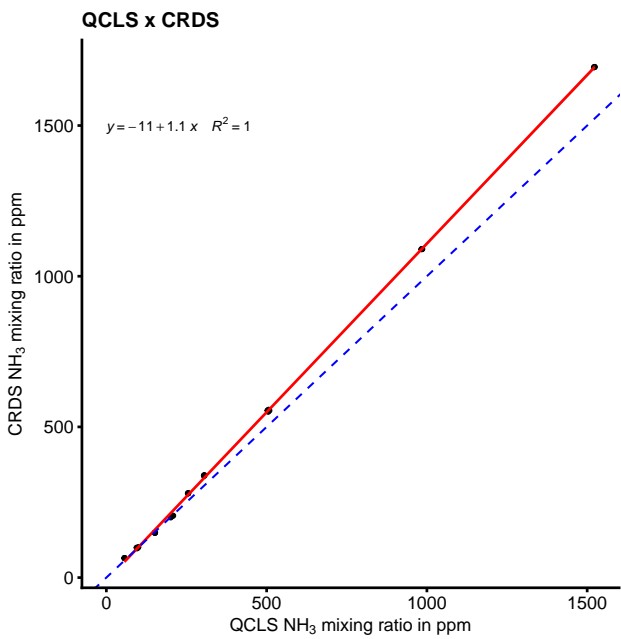

**Figure A2.** Linear regression plot between QCLS and CRDS $NH_3$ measurements for chapter 3.1

*Author contributions.* AK, MC and NB designed the experiments and AK and MC carried them out. AK wrote the first version of the manuscript, did the formal analysis and created the figures. MC installed and wrote the section on the laboratory setup, assisted with the QCLS measurements, and provided advice on the experimental design. NB gave advice on performing the experiment and the statistical

analysis, formulated the research idea, acquired the funding and helped with the planning and execution of the experiment. All authors reviewed and revised the manuscript.

*Competing interests.* The authors declare that they have no conflict of interest.

*Disclaimer.* The project was supported by funds of the German Federal Government's Special Purpose Fund held at Landwirtschaftliche Rentenbank under grant no. 894 820.

*Acknowledgements.* We want to thank Muhammad Humza for their contribution to the execution of the experiments described in this paper. Their assistance was instrumental in the completion of this research. We also appreciate the anonymous reviewers for their insightful comments and suggestions, which have helped enhance the quality of this manuscript.

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
