# Peer review of "Accuracy and sensitivity of NH3 measurements using the Dräger Tube Method"

_EGUsphere, 2024_

## Author Comment (AC2)

Dear referee RC2,

Thank you for your comprehensive review and constructive feedback on our manuscript. We will apply the suggestions you have made as best we can in the revision. Please find our responses to your specific comments below:

1. Comment: "To ensure that readers fully understand the technical details of the instrument setup and what each component is meant to be, I suggest rewording Lines 113-123 and improve the presentation of Figure 1."

   - Response: We will revise the technical details described in Section 2.1 to make the setup information as clear as possible to the reader. Fig. 1 will be revised to make it self-explanatory, as was also suggested by referee RC1.

2. Comment: "In my experience, calibration of NH3 instruments, even those based on laser spectroscopy, is challenging. The reading of NH3 usually takes long time to stabilize, which largely depends on pump flow rate, but low-frequency drift/variation still exists even under constant room temperature. I'm curious about the calibration of MIRO and Picarro analyzers and their performance during calibration, since it directly influences the results in Fig. 2."

   - Response: Yes, you are absolutely right, calibration of NH3 instruments based on laser spectroscopy is very challenging. Therefore, the manufacturer of the Picarro NH3 analyzer G2103 used in this study spends great effort in calibrating each delivered instrument already in the factory against a so-called "golden instrument" (S/N: AEDS2079), from which a specific calibration factor for each delivered instrument was derived in the factory. The reliability of this calibration factor was independently confirmed by ab initio calculations using the HITRAN2012 database (Rothman et al. 2013, Journal of Quantitative Spectroscopy and Radiative Transfer 130,4-50) and by the National Physical Laboratory of the United Kingdom (Martin et al. 2016, Applied Physics B 122, 1‐11). The stability of the Picarro CRDS analyzers was tested in a large intercomparison experiment with 47 CRDS analyzers, and a typical drift of about 0.1% slope/year was found (Yver Kwok et al., Atmos. Meas. Tech. 8, 3867‐3892). On the basis of these data, the manufacturer (Picarro) recommends that "There is no need to perform a true calibration in which the calibration slope is changed according to the results of a direct NH3 calibration experiment." Therefore, we chose the Picarro G2103 analyzer as the "master" in our experiment. Detailed information on the traceable calibration of Picarro NH3 analyzers can be found here: https://www.picarro.com/semiconductor/traceable_calibration_of_ammonia_nh3. We will add information on the traceable calibration of the Picarro analyzer to the revised version of the manuscript.

3. Comment: "According to the definitions of detection accuracy and precision (Lines 186-187), Line 205 should be written as "The detection accuracy and precision of the G2103, Drager Tubes and MGA[7] measurements are displayed in Fig. 2"."

   - Response: Fig. 2 does not show the detection accuracy and precision of the G2103 as we define it, since the G2103 was used as the reference for the Dräger Tubes and MGA[7]. Instead, we would prefer to rewrite the sentence as follows: "The detection accuracy and precision of the Dräger Tubes and MGA[7] measurements are displayed in Fig. 2."

4. Comment: "The authors suggest a more correct detection limit of 152-205 ppb for DTM measurement, but how to explain the significant drop in detection accuracy of DTM from 205 ppb to 305 ppb in Fig. 2?"

- Response: We are not quite sure why. What was different between the 205 ppb measurement and the 305 ppb measurement was the sampling date. 97, 152 and 205 ppb were measured later in March 2023 to find the detection limit of the Dräger tubes, while the other measurements were taken in October 2022. The temperature in the lab was about 5 °C colder in March, and the batch of Dräger Tubes used could also play a role in the quality of the readings. Another possibility, and probably the most likely, is observer error. It's possible that the Dräger Tubes are not sensitive enough to detect changes in the range of +- 100 ppb. NH3 concentrations are determined by the distance of the discoloration on the detector tubes. It is up to the observer to decide where exactly the discoloration stops. Sometimes the discoloration is only a very light blue, making observation even more difficult. We will discuss this in the revision.

5. Comment: "The detection precision is defined as the relative standard error of all measurements in Line 187, but in the caption of Fig. 2, it is claimed as the standard deviation. Please clarify."
   - Response: It should be the standard deviation. We will change line 187 accordingly.

6. Comment: "I don't agree with the statement in Lines 224-225. Fig. 3b clearly shows that the response time of the heated PTFE (4.73 min) is significantly shorter than both PU and Synflex tubing. By the way, check whether the figure 4.73 min is correct, it appears to be 6-7 min according on the y-axis scaling."
   - Response: While visually it appears to be significantly lower, we used the Kruskal-Wallis rank sum test for significant differences in Figure 3b. This test compares the medians rather than the means, while the figure shows the means. 4.73 min is incorrect, it should be 6.39 min. Thanks for spotting this error. We will correct the text and check for any other inconsistencies between the text and the figures shown.

7. Comment: "Line 281. The expression "A less sensitive detection limit …" is incorrect. It should be "A higher detection limit than originally assumed …" or "A lower measurement sensitivity than originally assumed …". Similarly, is it correct to say "the highest detection limit …" in Line 285?"
   - Response: We will use "A higher detection limit than originally assumed..." in line 281. We will also change the sentence in line 285 to "Therefore, instruments with high measurement sensitivity are preferred."

8. Comment: "The main conclusion of this paper is that the DTM is applicable only for large NH3 emission scenario due to its high detection limit of NH3 concentration. Hence, it would be helpful to give an estimate of flux detection limit for the dynamic chamber system with Drager tube used in this study, i.e. the lowest NH3 flux that the system can measure. In Section 4.1, the authors discussed two cases, in which DTM causes flux underestimation and overestimation. I prefer to have a discussion about the flux detection limit in this section."
   - Response: Our idea was to discuss the flux detection limit at first as well, but to determine fluxes with the calibrated DTM, one would first have to apply the calibration equation from Pacholski, 2006. There the NH3 fluxes depend strongly on the wind speed during the measurement time and are therefore variable. The underestimation of the daily NH3 flux depending on the wind speed would correspond to the daily flux detection limit. We could calculate the detection limit of the uncalibrated DTM. But the uncalibrated DTM is already known to underestimate real fluxes by one order of magnitude (Roelcke, 2002) and should not be used for quantitative measurements anyways. Calculating the flux detection limit of the uncalibrated DTM would in our opinion not add additional value to the discussion.

We will instead make it clear that we calculated a flux detection limit range for the calibrated DTM in the table and text.

9. Comment: "Lines 303-304. The NH3 fluxes disagree with those in Table 1."
   - Response: Thanks for spotting this, you are correct. We initially used the Beaufort scale range 0 to 6 for the information in the text, but later decided to include only the range 0 to 3 in the table because wind speeds above 4 m s-1 would be outside the range of the calibration function from Pacholski, 2006. We will update the text to be consistent with the table.

10. Comment: "Line 305. "daily relative error" means the error relative to the daily emission, but here the intended meaning of the author should be the flux error relative to the total NH3 volatilization for application rate of 60 kg N ha-1 and emission factor of 15%. Please rephrase the relevant content."
    - Response: You are correct. We will rephrase the sentence in line 305.

11. Comment: "Lines 308-311. The results in the two sentences are obscure, which makes it difficult to understand."
    - Response: We will rephrase the sentences to make this part clearer.

Response to technical corrections: We will make the suggested technical corrections to the text.